# Consistent Extended Kalman Filter-Based Cooperative Localization of Multiple Autonomous Underwater Vehicles

**DOI:** 10.3390/s22124563

**Published:** 2022-06-17

**Authors:** Fubin Zhang, Xingqi Wu, Peng Ma

**Affiliations:** 1School of Marine Science and Technology, Northwestern Polytechnical University, 127 West Youyi Road, Xi’an 710072, China; zhangfubin@nwpu.edu.cn; 2The 20th Research Institute of China Electronics Technology Group Corporation, 1 Baisha Road, Xi’an 710075, China; mapeng@mail.nwpu.edu.cn

**Keywords:** multiple AUVs, cooperative localization, extended Kalman filter, consistency

## Abstract

In order to solve the problem of inconsistent state estimation when multiple autonomous underwater vehicles (AUVs) are co-located, this paper proposes a method of multi-AUV co-location based on the consistent extended Kalman filter (EKF). Firstly, the dynamic model of cooperative positioning system follower AUV under two leaders alternately transmitting navigation information is established. Secondly, the observability of the standard linearization estimator based on the lead-follower multi-AUV cooperative positioning system is analyzed by comparing the subspace of the observable matrix of state estimation with that of an ideal observable matrix, it can be concluded that the estimation of state by standard EKF is inconsistent. Finally, aiming at the problem of inconsistent state estimation, a consistent EKF multi-AUV cooperative localization algorithm is designed. The algorithm corrects the linearized measurement values in the Jacobian matrix for cooperative positioning, ensuring that the linearized estimator can obtain accurate measurement values. The positioning results of the follower AUV under dead reckoning, standard EKF, and consistent EKF algorithms are simulated, analyzed, and compared with the real trajectory of the following AUV. The simulation results show that the follower AUV with a consistent EKF algorithm can keep synchronization with the leader AUV more stably.

## 1. Introduction

Due to the attenuation of underwater GPS signals and the complex marine environment, it is a challenge for autonomous underwater vehicles (AUV) to obtain high positioning accuracy [1]. Traditional acoustic-based positioning technologies, including long baseline (LBL), short baseline (SBL) and ultra-short baseline (USBL), are often limited by the operating area, acoustic beacon array arrangement, etc. [2]. The traditional approach for ocean-bottom monitoring is to deploy oceanographic sensors, record the data, and recover the instruments. This approach creates long lags in receiving the recorded information. In addition, if a failure occurs before recovery, all the data are lost. The most effective solution is established real-time communication between AUVs through underwater acoustic sensors. Underwater networks can also be used to increase the operation range of AUVs [3]. Therefore, multi-AUV cooperative positioning is put forward as a feasible method to improve the autonomous positioning accuracy of AUV [4,5]. In this paper, a lead-follower multi-AUV cooperative positioning system is adopted. The accuracy of the navigation sensor carried by the leader AUV is much higher than that of the follower AUV. Two leader AUVs broadcast their real-time positions alternately, and the follower AUV obtains the relative distance from the leader AUV through underwater acoustic sensors [6,7]. The underwater acoustic sensors equipped with the follower AUV alternately acquire the position coordinates of the leader AUV and the acoustic signal of the time of sending each signal. The leader AUV and the follower AUV are clock-synchronized, and one way travel time (OWTT) technology can be used to calculate the distance at different times. As the actual application environment is known, the sound speed is measured and fixed. If the time of signal transmission and reception and the speed of sound in the medium are known, the distance between the transmitter and the receiver can be calculated [8]. In this way, not only can the observability be maintained under the condition of infrequently changing formation, but the complexity of underwater acoustic communication can also be reduced [9,10]. In addition, each follower AUV independently receives the position information of the leader AUV and then infers its position according to the position information of the leader AUV. In this process, there is no information exchange between the follower AUVs [11]. The research results of this paper can be applied to any number of AUV teams because the actual formation rarely covers the range of 1000 m, so the attenuation of the signal is negligible [12]. No matter how many leader and follower AUVs are in the system, each follower AUV is independent, therefore all follower AUVs can obtain measurement information from the same single leader AUV at each sampling time step. The multi-AUV cooperative positioning system proposed in this paper can be applied in many aspects, such as improving ocean exploration, collecting oceanographic data, and ecological applications such as water quality and biological monitoring [13].

Many multi-AUV (or robot) cooperative localization algorithms have been proposed and successfully implemented in the literature, including standard-based extended Kalman filter (EKF) [14], particle filter (PF) [15], maximum a posteriori (MAP) [16], and moving horizon estimation (MHE) [17]. However, in these studies, the observability matrix of state estimation has a subspace with higher dimensions than the ideal observability matrix. The key problem of consistency has not been solved in these algorithms. According to the definition in [18], if the estimation error is zero mean, and the actual estimation error covariance (that is, the expected value of the square of the difference between the real state and the estimated state) is less than or equal to the updated state error covariance, then the state estimation of the dynamic system (such as EKF, UKF, and PF) is called consistent. Therefore, if the state estimation is inconsistent, it may lead to unreliable estimation and even divergence of estimation error. The inconsistency (over-trust) estimation problem of muti-AUV distributed cooperative positioning (that is, using the algorithm based on distributed EKF) is discussed, which is caused by data reuse and the correlation between AUVs, and an interleaved update (IU) algorithm for consistent cooperative positioning is proposed [19]. Compared with the above methods, Ref. [20] discussed the co-localization of isomorphic multi-robots, and proved that the mismatch between unobservable directions (for actual nonlinear systems and linearized systems) would lead to inconsistent estimation of global directions when using linearized estimators (such as EKF). The Jacobian matrix of state propagation was improved and the observability constraint was applied to the two algorithms. In order to improve the consistency, algorithm 1 needs extra storage space to store the last propagated state estimate, and algorithm 2 needs an additional variable which contains the running sum of all previous state corrections [21]. It shows that, for positioning and vision-assisted inertial navigation based on camera and inertial measurement unit (IMU), on the basis of observability analysis of the linearized system, one of the main sources of inconsistency is the false information obtained when the directional information is incorrectly projected, along with the direction corresponding to the rotation of the gravity vector [22]. To ensure that the observability of the proposed estimator matches that of the actual linear system, an observable constrained EKF algorithm was designed by modifying the state propagation and measurement Jacobian matrix. It is consistent with the methods of [20,21].

Underwater acoustic communication between AUVs is always limited by large delay and low bandwidth. Many kinds of research on homogeneous multi-robots, such as Huang’s work, are not suitable for underwater scenes. The lead-follower multi-AUV cooperative positioning discussed in this paper is less limited by underwater acoustic communication and more suitable for the underwater environments. Based on observability analysis, the measurement Jacobian matrix is obtained under the constraint of relative position. A consistent linearization estimator is designed for cooperative localization. If the motion conditions and alternating communication means are met in this paper, the proposed state estimator can also be applied to cooperative navigation in other situations, such as unmanned aerial vehicles and robots.

To sum up, the main contributions of this work are as follows.

(1)In this paper, the cooperative localization of heterogeneous AUVs based on underwater acoustic communication is studied. In order to improve the positioning accuracy of the follower AUV, the follower AUV can only alternately obtain the relative range measurement values with the two leading AUVs through OWTT, thus reducing the complexity of acoustic communication. This can result not only in better observability than the single leader with the same communication load, but also in avoiding changing the formation frequently. In addition, the research results can be easily extended to any number of AUV teams and the acoustic communication load will not increase.(2)According to different distance measurement information, the observability matrix of the whole system is decomposed into two independent parts. The whole positioning system is observable, two state estimation and decomposition subsystems related to a single leader are observable, and two ideal decomposition subsystems related to a single leader are not observable. As the rank of observability matrix of decomposition subsystem calculated by standard linearization estimator (such as EKF) is larger than that calculated by the ideal state value, it will lead to inconsistent estimation of the position state of follower AUV.(3)In order to improve the consistency of state estimation, this paper designs a consistent EKF algorithm for multi-AUV cooperative positioning. As the ideal state value cannot be used to calculate the Jacobian matrix, in order to improve the consistency of the standard linearization estimator, the algorithm uses the designed initial zero space vector related to the relative position to construct the constrain conditions of each recursive time step, and then obtain the modified measurement Jacobian matrix under the constraint conditions and prove that the state propagation Jacobian matrix is not affected by the initial zero-space vector.

The rest of this paper is organized as follows. Section 2 describes the formulation of a discrete-time nonlinear model of the cooperative positioning system and the corresponding standard EKF algorithm. In Section 3, the observable matrix is constructed and the inconsistency of the standard linearized system is analyzed. In Section 4, a consistent algorithm based on EKF is proposed. Section 5 gives a series of numerical simulation and analysis results, to verify the performance of the algorithm. Finally, in Section 6, the conclusions and future research directions are drawn.

## 2. Theoretical Basis of Multi-AUV Cooperative Positioning

In this multi-AUV cooperative positioning system, all AUV clocks are synchronized before transmission, as shown in Figure 1. In the process of formation navigation, follower AUV can alternately obtain position information from two leader AUVs. For example, the leader AUV_1_ starts broadcasting its position at $t_{1}$ time, and leader AUV_2_ starts broadcasting its position at $t_{2}$ time, and the time interval between two leader AUVs broadcasting their positions is the same. A follower AUV can alternately obtain relative distance measurements from two leader AUVs through the OWTT characteristics of acoustic broadcast [23]. As one of the most commonly used co-location filtering algorithms, the standard EKF can make full use of the statistical information of measurement equation and measurement error to recursively solve the follower AUV state estimation based on the linearization of the nonlinear co-location system model. Moreover, the algorithm is simple to implement and the estimation accuracy is high. Generally speaking, the standard EKF state estimator is divided into two steps, as follows.

### 2.1. Motion State Prediction

As the actual depth information can be directly measured by the pressure sensor in real-time, the depth does not need to be considered in the system equations and the practical working environment of an AUV can be simplified to a two-dimensional (2D) space [24]. In the local level coordinate system, the two-dimensional state (2D) vector follower the AUV at time *k* is $X_{k}={\left[\begin{array}{cc} p_{k}^{T} & \phi_{k} \end{array}\right]}^{T}$, in which $p_{k}={\left[\begin{array}{cc} x_{k} & y_{k} \end{array}\right]}^{T}$ is the location, $x_{k}$ and $y_{k}$ follow the east and north positions of AUV, respectively, and $\phi_{k}$ is the heading angle. The kinematic equation follower AUV can be expressed by a nonlinear discrete-time system:(1)$$X_{k+1}=f\left(X_{k},u_{k},\omega_{k}\right)=\left[\begin{array}{c} x_{k}+\delta t\cdot v_{k}\cdot \cos\left(\phi_{k}\right)\\ y_{k}+\delta t\cdot v_{k}\cdot \sin\left(\phi_{k}\right)\\ \phi_{k}+\delta t\cdot \omega_{k} \end{array}\right]$$

Equation (Equation 1) is the AUV motion model under the ideal condition, δt is a constant sampling time interval. Assuming that the measured input of the sensor in the actual model is interfered with by Gaussian white noise, the measured input, real input, and sensor noise are, respectively:(2)$$u_{mk}=\left[\begin{array}{c} v_{mk}\\ \omega_{mk} \end{array}\right],u_{k}=\left[\begin{array}{c} v_{k}\\ \omega_{k} \end{array}\right],w_{k}=\left[\begin{array}{c} w_{vk}\\ w_{\phi k} \end{array}\right]$$

$u_{k}=u_{mk}+w_{k}$. The noise covariance matrix is:(3)$$Q_{k}=E\left[w_{k}w_{k}^{T}\right]=\left[\begin{array}{cc} \sigma_{v,k}^{2} & 0\\ 0 & \sigma_{\phi ,k}^{2} \end{array}\right]$$

We adopted ${\hat{X}}_{k-1}$ as the state estimate; the predicted state estimate ${\hat{X}}_{k/k-1}$ at time step *k* can then be expressed via the kinematic Equation (Equation 1). We consider ${\hat{F}}_{k/k-1}$ and ${\hat{G}}_{k/k-1}$ to be Jacobian matrices (i.e., system matrices for the linearized system) for $f\left({\hat{X}}_{k-1},u_{k-1}\right)$, with respect to ${\hat{X}}_{k-1}$ and ${\hat{u}}_{k-1}$, respectively. These can be expressed as:(4)$${\hat{F}}_{k-1}=\frac{\partial f}{\partial {\hat{X}}_{k-1}}=\left[\begin{array}{ccc} 1 & 0 & -\delta t\cdot v_{k-1}\cdot \sin\left({\hat{\phi }}_{k-1}\right)\\ 0 & 1 & \delta t\cdot v_{k-1}\cdot \cos\left({\hat{\phi }}_{k-1}\right)\\ 0 & 0 & 1 \end{array}\right]$$
(5)$${\hat{G}}_{k-1}=\frac{\partial f}{\partial u_{k-1}}=\left[\begin{array}{cc} \delta t\cdot \cos\left({\hat{\phi }}_{k-1}\right) & 0\\ \delta t\cdot \sin\left({\hat{\phi }}_{k-1}\right) & 0\\ 0 & \delta t \end{array}\right]$$

In the process of state estimation using EKF, the predicted state covariance matrix ${\hat{P}}_{k/k-1}$ can then be computed as:(6)$${\hat{P}}_{k/k-1}={\hat{F}}_{k-1}{\hat{P}}_{k-1}{\hat{F}}_{k-1}^{T}+{\hat{G}}_{k-1}^{T}$$

### 2.2. Measurement Update Model

We consider $X_{i,k}={\left[\begin{array}{cc} p_{i,k}^{T} & \phi_{i,k} \end{array}\right]}^{T}={\left[\begin{array}{ccc} x_{i,k} & y_{i,k} & \phi_{i,k} \end{array}\right]}^{T}\left(i=1,2\right)$ to be the state vector for leader AUV_*i*_. The control inputs for all leader and follower AUVs are equal and fixed (i.e., $u_{i,k}=u_{k}$) to maintain motion formation. In the presence of acoustic range-only measurements, the measured range model at time step *k* can be expressed as:(7)$$Z_{i,k}=h\left(X_{i,k},X_{k}\right)+\upsilon_{i,k}=∥p_{i,k}-p_{k}∥+\upsilon_{i,k}$$
where $d_{i,k}=∥p_{i,k}-p_{k}∥$ is the Euclidean distance between the positions of leader AUV _*i*_ and follower AUV. The term $\upsilon_{i,k}$ is the range measurement noise following the Gaussian distribution $N\left(0,R_{i}\right)$. The index $i=\lambda \left(k\right)\in \left\{1,2\right\}$ will alternate with time; when $k=2\gamma \left(\gamma \in N^{+}\right)$, we set $\lambda \left(k\right)=2$, otherwise $\lambda \left(k\right)=1$.

By linearizing Equation (Equation 7) with first-order Taylor expansion, the Jacobian matrix of the measurement model can be obtained as follows:(8)$${\hat{H}}_{i,k}=\frac{\partial h}{\partial X}{|}_{{\hat{X}}_{k/k-1}}=\left[\begin{array}{cc} -\frac{{\left(p_{i,k}-{\hat{p}}_{k/k-1}\right)}^{T}}{∥p_{i,k}-{\hat{p}}_{k/k-1}∥} & 0 \end{array}\right]$$

Subsequently, we employ a direct range measurement (Equation (Equation 7)) to update the EKF and correct the accumulated dead-reckoning errors for follower AUVs. The residual measurement between measured and predicted ranges and the Kalman gain can be calculated as follows:(9)$$r_{i,k}=Z_{i,k}-h\left(X_{i,k},{\hat{X}}_{k/k-1}\right)$$
(10)$$K_{k}={\hat{P}}_{k/k-1}{\hat{H}}_{i,k}^{T}{\left({\hat{H}}_{i,k}{\hat{P}}_{k/k-1}{\hat{H}}_{i,k}^{T}+R_{i,k}\right)}^{-1}$$

Using Equations (Equation 9) and (Equation 10), the state estimation and covariance are updated by distance measurement information can be obtained as follows:(11)$${\hat{X}}_{k}={\hat{X}}_{k/k-1}+K_{k}r_{i,k}$$
(12)$${\hat{P}}_{k}=\left(I-K_{k}{\hat{H}}_{i,k}\right){\hat{P}}_{k/k-1}$$

## 3. Observability and Consistency Analysis of Multi-AUV Cooperative Positioning

Traditionally, system observability is determined by whether the state of a system can be determined from the output (and input) measurements. If the initial state of a system can be uniquely determined for any time in a finite interval, the system is observable, otherwise it is not observable [25]. Thus, if a cooperative localization system is observable, follower AUVs will be localizable. Based on this observability analysis, we describe the influence of observability properties on standard EKF consistency.

A local observability matrix [26] can be adopted for linearized time-varying systems to analyze observability by computing rank conditions. If the observability matrix is full rank (i.e., the rank of the observability matrix equals the dimension of the system state), the linearized time-varying system is locally weakly observable. This indicates the matrix is observable in one local time interval but does not mean it exhibits observability in every time interval. The observability matrix of cooperative localization can be decomposed into two corresponding components [27], according to exteroceptive measurement information alternately acquired from two leader AUVs. For a linearized time-varying systems, the observable matrix consists of state transition matrix and measurement Jacobian matrix. The state transition matrix and measurement Jacobian matrix are calculated at the selected linearization point. In other words, the observability matrix is a function of the linearization point. Therefore, the choice of linearization point will affect the observability of linearized time-varying system, which is the key fact that the comparison between ideal state value and state estimation value is the basis for analyzing observability. Generally speaking, although it is impossible to calculate the Jacobian matrix with the ideal state value, the linearization point should still be as close as possible to the ideal state value.

This article will be in the time interval $\left[1,\;k\right]$ ($k=2\gamma ,\;\gamma \in N^{+}$), the observable matrix based on state estimation is constructed as follows:(13)$$\hat{O}=\left[\begin{array}{c} {\hat{H}}_{1,1}\\ {\hat{H}}_{2,2}{\hat{F}}_{1}\\ \vdots \\ {\hat{H}}_{1,k-1}{\hat{F}}_{k-2}\cdots {\hat{F}}_{1}\\ {\hat{H}}_{2,k}{\hat{F}}_{k-1}\cdots {\hat{F}}_{1} \end{array}\right]=\underset{{\hat{O}}_{1}}{\underset{︸}{\left[\begin{array}{c} {\hat{H}}_{1,1}\\ 0\\ \vdots \\ {\hat{H}}_{1,k-1}{\hat{F}}_{k-2}\cdots {\hat{F}}_{1}\\ 0 \end{array}\right]}}+\underset{{\hat{O}}_{2}}{\underset{︸}{\left[\begin{array}{c} 0\\ {\hat{H}}_{2,2}{\hat{F}}_{1}\\ \vdots \\ 0\\ {\hat{H}}_{2,k}{\hat{F}}_{k-1}\cdots {\hat{F}}_{1} \end{array}\right]}}$$

As such, it is not difficult to determine the submatrices ${\hat{O}}_{1}$ and ${\hat{O}}_{2}$ constructed by decomposing the observability matrix $\hat{O}$. In Equations (Equation 4) and (Equation 8), we observe that ${\hat{H}}_{i,k}$ and ${\hat{F}}_{k-1}$ are related to the information broadcasted by the leader AUV_*i*_ only. This demonstrates the measurement information of submatrices ${\hat{O}}_{i}$ results only from the leader AUV ${}_{i}$.

In the case that the leader AUV remains to maneuver, the control input of the vehicle $\upsilon_{k}\neq 0$. We consider $p_{k}^{T}=\left[\begin{array}{cc} x_{k} & y_{k} \end{array}\right]$ to be the state vector for the follower AUV. To simplify this analysis, we substitute Equation (Equation 14) and (Equation 15) into Equation (Equation 4) to rearrange the Jacobian matrices ${\hat{F}}_{k-1}$ and ${\hat{H}}_{i,k}$ equivalently, as follows:(14)$${\hat{x}}_{k/k-1}-{\hat{x}}_{k-1}=\delta t\cdot v_{k-1}\cdot \cos\left({\hat{\phi }}_{k-1}\right)$$
(15)$${\hat{y}}_{k/k-1}-{\hat{y}}_{k-1}=\delta t\cdot v_{k-1}\cdot \sin\left({\hat{\phi }}_{k-1}\right)$$
(16)$${\hat{F}}_{k-1}=\left[\begin{array}{cc} I_{2} & \left[\begin{array}{c} -{\hat{y}}_{k/k-1}-{\hat{y}}_{k-1}\\ {\hat{x}}_{k/k-1}-{\hat{x}}_{k-1} \end{array}\right]\\ 0_{1\times 2} & 1 \end{array}\right]=\left[\begin{array}{cc} I_{2} & C({\hat{p}}_{k/k-1}-{\hat{p}}_{k-1})\\ 0_{1\times 2} & 1 \end{array}\right]$$
(17)$${\hat{H}}_{i,k}=-{\hat{d}}_{i,k}^{-1}\left[\begin{array}{cc} {\left(p_{i,k}-{\hat{p}}_{k/k-1}\right)}^{T} & 0 \end{array}\right]$$
where $C=\left[\begin{array}{cc} 0 & -1\\ 1 & 0 \end{array}\right]$, ${\hat{d}}_{i,k}=∥p_{i,k}-{\hat{p}}_{k/k-1}∥.$

Furthermore, we define $\delta {\hat{p}}_{s}={\hat{p}}_{s}-{\hat{p}}_{s/s-1}$, which are not equal to 0 in practice. The following expression can then be derived using (Equation 16) and (Equation 17):(18)$${\hat{H}}_{1,1}=-{\hat{d}}_{1,1}^{-1}\left[\begin{array}{cc} {\left(p_{1,1}-{\hat{p}}_{1/0}\right)}^{T} & 0 \end{array}\right]=-{\hat{d}}_{1,1}^{-1}\left[\begin{array}{cc} {\left(p_{1,1}-{\hat{p}}_{1}\right)}^{T} & 0 \end{array}\right]$$
(19)$$\begin{array}{cc} {\hat{H}}_{i,k}{\hat{F}}_{k-1}\cdots {\hat{F}}_{1} & =-{\hat{d}}_{i,k}^{-1}\left[\begin{array}{cc} {\left(p_{i,k}-{\hat{p}}_{k/k-1}\right)}^{T} & {\left(p_{i,k}-{\hat{p}}_{k/k-1}\right)}^{T}C\left({\hat{p}}_{k/k-1}-{\hat{p}}_{k-1}+{\hat{p}}_{k-1/k-2}-\cdots -{\hat{p}}_{1}\right) \end{array}\right]\\ \; & =-{\hat{d}}_{i,k}^{-1}\left[\begin{array}{cc} {\left(p_{i,k}-{\hat{p}}_{k}+\delta {\hat{p}}_{k}\right)}^{T} & {\left(p_{i,k}-{\hat{p}}_{k}+\delta {\hat{p}}_{k}\right)}^{T}C\left({\hat{p}}_{k/k-1}-\overset{k-1}{\underset{s=2}{\Sigma }}\delta {\hat{p}}_{s}-{\hat{p}}_{1}\right) \end{array}\right] \end{array}$$
${\hat{p}}_{1}={\hat{p}}_{1/0}$ is the follower AUV initial position estimate. Then, by substituting (Equation 18) and (Equation 19) into the linearized observable matrix (Equation 13) at the same time, it can be proved by determinant transformation that:(20)$$rank\left(\hat{O}\right)=rank\left({\hat{O}}_{1}\right)=rank\left({\hat{O}}_{2}\right)=3$$

Therefore, the observability matrix $\hat{O}$ and ${\hat{O}}_{i}$ are full rank and the cooperative positioning system is observable, which ensures that the follower AUV state can be solved by a linearization estimator (such as EKF).

Next, the observability of the cooperative position system is analyzed using the ideal state value instead of the estimated value, in this description:(21)$$X_{k}={\hat{X}}_{k}={\hat{X}}_{k/k-1}\;p_{k}={\hat{p}}_{k}={\hat{p}}_{k/k-1}$$

By substituting Equation (Equation 21) into Equation (Equation 13), a linearized observability matrix based on the ideal state values can be obtained.
(22)$$O=\left[\begin{array}{c} H_{1,1}\\ H_{2,2}F_{1}\\ \vdots \\ H_{1,k-1}F_{k-2}\cdots F_{1}\\ H_{2,k}F_{k-1}\cdots F_{1} \end{array}\right]=\underset{O_{1}}{\underset{︸}{\left[\begin{array}{c} H_{1,1}\\ 0\\ \vdots \\ H_{1,k-1}F_{k-2}\cdots F_{1}\\ 0 \end{array}\right]}}+\underset{O_{2}}{\underset{︸}{\left[\begin{array}{c} 0\\ H_{2,2}F_{1}\\ \vdots \\ 0\\ H_{2,k}F_{k-1}\cdots F_{1} \end{array}\right]}}$$

Its corresponding determinant matrix becomes:(23)$$H_{1,1}=-d_{1,1}^{-1}\left[\begin{array}{cc} {\left(p_{1,1}-p_{1}\right)}^{T} & 0 \end{array}\right]$$
(24)$$H_{i,k}F_{k-1}\cdots F_{1}=-d_{i,k}^{-1}\left[\begin{array}{cc} {\left(p_{i,k}-p_{k}\right)}^{T} & {\left(p_{i,k}-p_{k}\right)}^{T}C\left(p_{k}-p_{1}\right) \end{array}\right]\;,\;k\geq 2$$

By substituting Equations (Equation 23) and (Equation 24) into the observability matrix (Equation (Equation 22)), we denote the submatrices $O_{1}=\left[\begin{array}{ccc} m_{1} & m_{2} & m_{3} \end{array}\right]$ and $O_{2}=\left[\begin{array}{ccc} n_{1} & n_{2} & n_{3} \end{array}\right]$. While keeping the motion formation constant (i.e., $u_{i,k}=u_{k}$), the submatrix column vectors will remain $m_{1}=\beta_{1}m_{2}$ and $n_{1}=\beta_{2}n_{2}$, in which $\beta_{1}$ and $\beta_{2}$ are related to the relative positions between leader AUV_*i*_ and follower AUVs [28]. At this point, the observability matrix is full rank, but submatrix $O_{1}$ and $O_{2}$ are not full rank, therefore:(25)$$rank\left(O\right)=3\;rank\left(O_{1}\right)=rank\left(O_{2}\right)=2$$

Note: In a large underwater task region, requiring all AUVs to maintain a constant motion formation along the same linear direction is an effective planning strategy to ensure full-region coverage. In addition, due to the sea water resistance, frequently changing the motion formation of AUVs will accelerate energy consumption and require additional time.

In comparing the rank expressions (Equation 20) and (Equation 25), it is evident the submatrices ${\hat{O}}_{1}$ and ${\hat{O}}_{2}$ in the linearized estimator have observable subspaces of higher dimension than those of $O_{1}$ and $O_{2}$, which are calculated using ideal state values. As a result, the linearized estimator acquires nonexistent and spurious information alternating along the varied unobservable directions from each leader AUV range measurement. This can lead to inconsistent estimation, smaller uncertainties, and larger errors [29]. To solve this problem, we propose a consistent estimation algorithm for cooperative multiple-AUV localization as described in the following section.

## 4. Consistent EKF Algorithm for Multi-AUV Cooperative Positioning

In practice, it is impossible to acquire ideal state values with noise and errors in the measurement system. As such, we cannot calculate Jacobian matrices using ideal state values, as opposed to the latest state estimate values in a standard linearized estimator. Thus, in this paper, by modifying the state transition matrix and measuring the Jacobian matrix, the observability matching between the actual linearized system and the real system is ensured.

### 4.1. Zero-Space Vector

To ensure that the rank of the observability matrices is consistent with that of the real state value when calculating the state transition matrix and measuring jacobian matrix with the state estimate value, the observability constraint ${\hat{O}}_{i}{\hat{N}}_{i,1}=0$ can be added to achieve the purpose that the matrix ${\hat{O}}_{1}$ and ${\hat{O}}_{2}$ are non-full rank [20,21]. This can be summarized as:(26)$$\left\{\begin{array}{c} {\hat{H}}_{1,1}{\hat{N}}_{1,1}=0,\;\forall k=1\\ {\hat{H}}_{2,2}{\hat{F}}_{1}{\hat{N}}_{2,1}=0,\;\forall k=2\\ \vdots \\ {\hat{H}}_{i,k}{\hat{F}}_{k-1}\cdots {\hat{F}}_{1}{\hat{N}}_{i,1}=0,\;\forall k>2 \end{array}\right.$$
where ${\hat{N}}_{i,1}$ is a design choice used to control the observable subspace of submatrices ${\hat{O}}_{i}$, which is the zero-space vector designed by using the initial state estimation value and can be computed analytically using:(27)$$\left\{\begin{array}{c} {\hat{N}}_{i,1}={\left[\begin{array}{ccc} 1 & -{\hat{\beta }}_{i,1} & 0 \end{array}\right]}^{T}\\ {\hat{\beta }}_{i,1}=\frac{\left(x_{i,1}-{\hat{x}}_{1}\right)}{\left(y_{i,1}-{\hat{y}}_{1}\right)} \end{array}\right.$$

According to the constraint expressions (Equation 26) and (Equation 27) described above, we can further define the following recursive expressions as in [30]:(28)$${\hat{N}}_{i,k}={\hat{F}}_{k-1}\cdots {\hat{F}}_{1}{\hat{N}}_{i,1}\;\forall k\geq 2$$
(29)$${\hat{N}}_{i,k}={\left[\begin{array}{ccc} 1 & -{\hat{\beta }}_{i,k} & 0 \end{array}\right]}^{T}$$
where ${\hat{N}}_{i,k}$ is a design function with respect to the state estimate values.

With the definitions provided in (Equation 28), the constraint conditions (Equation 26) can be equivalently satisfied by modifying the Jacobian matrices at each time step, such that:(30)$$\left\{\begin{array}{c} {\hat{F}}_{k-1}{\hat{F}}_{k-2}{\hat{N}}_{i,k-2}={\hat{N}}_{i,k}\\ {\hat{H}}_{i,k}{\hat{N}}_{i,k}=0 \end{array}\right.,\;\forall k\in {ℜ}_{i}$$
where ${ℜ}_{i}\left(i=1,2\right)$ represents the set of indices for sample times with measurement information from only leader AUV${}_{i}$, respectively. Alternating communication refers to information between leader and follower AUVs, such as ${ℜ}_{1}=\left\{1,3,\cdots ,k-1\right\}$ and ${ℜ}_{2}=\left\{2,4,\cdots ,k\right\}$.

### 4.2. Modification of Jacobian Matrices

In computing each covariance prediction (Equation 6), we must ensure the constraint condition ${\hat{F}}_{k-1}{\hat{F}}_{k-2}{\hat{N}}_{i,k-2}={\hat{N}}_{i,k}$ is satisfied. We define ${\hat{\Phi }}_{k}^{\left(13\right)}$ and ${\hat{\Phi }}_{k}^{\left(23\right)}$ to be unknown elements of the Jacobian matrix ${\hat{F}}_{k}$, ${\hat{\Phi }}_{k}^{\left(13\right)}=-\delta t\cdot v_{k-1}\cdot \cos\left({\hat{\phi }}_{k-1}\right)$, ${\hat{\Phi }}_{k}^{\left(23\right)}=\delta t\cdot v_{k-1}\cdot \sin\left({\hat{\phi }}_{k-1}\right)$. The matrix ${\hat{F}}_{k}$ is then reconstructed in the basic row-column structure as follows:(31)$${\hat{F}}_{k}=\left[\begin{array}{ccc} 1 & 0 & {\hat{\Phi }}_{k}^{\left(13\right)}\\ 0 & 1 & {\hat{\Phi }}_{k}^{\left(23\right)}\\ 0 & 0 & 1 \end{array}\right]$$

From this relationship (Equation (Equation 30)), the following expression can be derived by substituting Equations (Equation 29) and (Equation 31) into the constraint equation ${\hat{F}}_{k-1}{\hat{F}}_{k-2}{\hat{N}}_{i,k-2}={\hat{N}}_{i,k}$:(32)$$\begin{array}{c} \left[\begin{array}{ccc} 1 & 0 & {\hat{\Phi }}_{k-1}^{\left(13\right)}\\ 0 & 1 & {\hat{\Phi }}_{k-1}^{\left(23\right)}\\ 0 & 0 & 1 \end{array}\right]\left[\begin{array}{ccc} 1 & 0 & {\hat{\Phi }}_{k-2}^{\left(13\right)}\\ 0 & 1 & {\hat{\Phi }}_{k-2}^{\left(23\right)}\\ 0 & 0 & 1 \end{array}\right]\left[\begin{array}{c} 1\\ -{\hat{\beta }}_{i,k-2}\\ 0 \end{array}\right]\\ =\left[\begin{array}{ccc} 1 & 0 & {\hat{\Phi }}_{k-1}^{13}+{\hat{\Phi }}_{k-2}^{\left(13\right)}\\ 0 & 1 & {\hat{\Phi }}_{k-1}^{23}+{\hat{\Phi }}_{k-2}^{\left(23\right)}\\ 0 & 0 & 1 \end{array}\right]\left[\begin{array}{c} 1\\ -{\hat{\beta }}_{i,k-2}\\ 0 \end{array}\right]=\left[\begin{array}{c} 1\\ -{\hat{\beta }}_{i,k-2}\\ 0 \end{array}\right] \end{array}$$
(33)$$\left[\begin{array}{ccc} 1 & 0 & {\hat{\Phi }}_{k-1}^{\left(13\right)}\\ 0 & 1 & {\hat{\Phi }}_{k-1}^{\left(23\right)}\\ 0 & 0 & 1 \end{array}\right]\left[\begin{array}{ccc} 1 & 0 & {\hat{\Phi }}_{k-2}^{\left(13\right)}\\ 0 & 1 & {\hat{\Phi }}_{k-2}^{\left(23\right)}\\ 0 & 0 & 1 \end{array}\right]\left[\begin{array}{c} 1\\ -{\hat{\beta }}_{i,k-2}\\ 0 \end{array}\right]=\left[\begin{array}{c} 1\\ -{\hat{\beta }}_{i,k}\\ 0 \end{array}\right]\Rightarrow \;{\hat{\beta }}_{i,k}={\hat{\beta }}_{i,k-2}$$

We can further determine:(34)$${\hat{N}}_{i,k}={\hat{N}}_{i,1}={\left[\begin{array}{ccc} 1 & -\frac{\left(x_{i,1}-{\hat{x}}_{1}\right)}{\left(y_{i,1}-{\hat{y}}_{1}\right)} & 0 \end{array}\right]}^{T}$$

Finally, the constraint expression ${\hat{H}}_{i,k}{\hat{N}}_{i,k}=0$ can equivalently be replaced by:(35)$${\hat{H}}_{i,k}{\hat{N}}_{i,1}=0,\;\forall k\in {ℜ}_{i}$$

In expressions (Equation 34) and (Equation 35), we know ${\hat{H}}_{i,k}$ is an unknown time-varying vector and ${\hat{N}}_{i,1}$ is a fixed-quantity zero-space vector. To fulfil the constraints in Equation (Equation 35) and obtain a modified measurement Jacobian matrix, we solve the following minimization problem:(36)$$\min_{{\hat{H}}_{i,k}^{*}}\;{∥{\hat{H}}_{i,k}^{*}-{\hat{H}}_{i,k}∥}_{F}^{2}\;\mathrm{subject}\;\mathrm{to}\;{\hat{H}}_{i,k}^{*}{\hat{N}}_{i,1}=0$$
where ${∥\;∥}_{F}^{2}$ denotes the Frobenius matrix norm. After employing the method of Lagrange multipliers and analytically solving the corresponding Karush–Kuhn–Tucker (KKT) optimality conditions, the optimal solution to the minimization problem described by Equation (Equation 36) can be expressed as:(37)$${\hat{H}}_{i,k}^{*}={\hat{H}}_{i,k}-{\hat{H}}_{i,k}{\hat{N}}_{i,1}{\left({\hat{N}}_{i,1}^{T}{\hat{N}}_{i,1}\right)}^{-1}{\hat{N}}_{i,1}^{T}$$

Notice the zero-space vector ${\hat{N}}_{i,k}$ is relevant to the geometric configuration between the leader and follower AUVs. In other words, observability-constrained conditions are affected by the relative position configurations between AUVs. From this perspective, the optimal solution ${\hat{H}}_{i,k}^{*}$ can be considered a modified measurement Jacobian matrix under relative position constraints.

In the previous sections, we presented only a consistent estimator for 2D linearized cooperative localization systems, which is related to observability properties and relative position configurations. In contrast to this 2D system, the position relationships between AUVs can be expanded to spatial structures with depth information in three-dimensional (3D) systems. The depth information of the follower AUV can be obtained by using the depth sensor, and its projection on the leader AUV can be calculated using the Pythagorean theorem in 3D systems. This method turns a 3D system into a 2D system. Therefore, a similar analytic method can still be applied in the design of consistent estimators for 3D linearized systems.

## 5. Simulation Results

In this section, a series of simulation results will be given to illustrate the effectiveness of the proposed algorithm. In this paper, it is assumed that follower AUV can alternately obtain the distance between two leader AUVs. As acoustic signals cannot carry too much information, it is necessary to minimize the communication frequency under the premise of a stable system. In [31], the communication frequency of multi-AUV cooperative navigation is selected as 1 Hz. In the process of multi-AUV cooperative work, besides cooperative navigation information between AUVs, some formation control commands need to be transmitted, and the communication rate decreases with the increase in distance between communication devices. Therefore, it is feasible to assume that the update frequency of distance measurement is 0.2 Hz and the covariance is $R_{1}=R_{2}$ = (2 m)${}^{2}$. Control inputs were the same for all AUVs, keeping the navigation formation constant, setting the constant forward speed as $\nu_{k}=4$ m/s. When all AUVs move along a straight line, the angular velocity is $\omega_{k}=0$. When turning, the angular velocity is ±0.015 rad/s, as shown in Figure 2. In addition, we set the follower AUV covariance of control input to $Q=diag\left[\begin{array}{cc} {\left(0.5\;m/s\right)}^{2} & {\left(0.001\;\mathrm{rad}/s\right)}^{2} \end{array}\right]$. In the two-dimensional rectangular coordinate system, the initial position of the follower AUV is (500, 500), and the initial positions of the two leader AUVs are (1000, 382) and (1000, 636), respectively. Therefore, the initial zero-space vector can be obtained with ${\hat{N}}_{1,1}=\left[\begin{array}{ccc} 1 & 3.9 & 0 \end{array}\right]$ and ${\hat{N}}_{2,1}=\left[\begin{array}{ccc} 1 & -3.7 & 0 \end{array}\right]$.

Localization results from three different algorithms (dead-reckoning, standard EKF, and consistent EKF) are presented to assess the performance of the proposed consistent EKF algorithm and provide a comparison with the true follower AUV trajectory shown in Figure 3. Position information and acoustic range measurements from the leader AUV demonstrate the cooperative localization trajectories of standard EKF and consistent EKF, including bounded errors. The dead-reckoning (DR) error divergence for follower AUVs has been effectively overcome and the estimated trajectory of consistent EKF is superior to standard EKF. Furthermore, as the relative directions between leader and follower AUVs are time-invariant, the observability of cooperative localization systems does not change, and the proposed consistent algorithm is still available in the case of turning.

To further demonstrate the advantages of the proposed consistent EKF algorithm, the root-mean-square-error (RMSE) was calculated for two different localization algorithms. The position and heading of the follower AUV at k time RMSE was determined using 100 Monte Carlo simulations:(38)$$p_{k}^{R}=\sqrt{\frac{1}{100}\sum_{l=1}^{100}{\left({\hat{p}}_{k,l}-p_{k,l}\right)}^{T}\left({\hat{p}}_{k,l}-p_{k,l}\right)}$$
(39)$$\varphi_{k}^{R}=\sqrt{\frac{1}{100}\sum_{l=1}^{100}{\left({\hat{\varphi }}_{k,l}-\phi_{k,l}\right)}^{T}\left({\hat{\varphi }}_{k,l}-\varphi_{k,l}\right)}$$

Through the comparison of Figure 4 and Figure 5, it is found that the RMSE of cooperative positioning based on the consistent EKF algorithm is lower than that of the standard EKF, which demonstrates the proposed consistent EKF algorithm is more suitable for cooperative localization based on multi-AUV under alternate navigation. This is partly because relative position constraints were introduced into the consistent EKF through observability-constrained conditions.

The normalized estimation error squared (NEES) is the most common criterion for evaluating the consistency of state estimators for dynamic systems. Specifically, the NEES of an N-dimensional Gaussian random variable follows a $\chi^{2}$ distribution with N degrees of freedom [32]. If the designed cooperative localization algorithm for state estimation (i.e., position and heading angle) of a follower AUV is consistent, the NEES expected values for position and heading angle estimates will be close to 2 and 1, respectively. In other words, expected values which are closer to actual NEES estimations indicate better consistency for dynamic system state estimators. The red dashed lines in Figure 6 and Figure 7 represent NEES expected values. It is evident that the consistency of the proposed consistent EKF algorithm is significantly higher than that of standard EKF.
(40)$$p_{k}^{N}=\frac{1}{100}\sum_{l=1}^{100}{\left({\hat{p}}_{k,l}-p_{k,l}\right)}^{T}P_{1,k,l}^{-1}\left({\hat{p}}_{k,l}-p_{k,l}\right)$$
(41)$$\varphi_{k}^{N}=\frac{1}{100}\sum_{l=1}^{100}{\left({\hat{\varphi }}_{k,l}-\varphi_{k,l}\right)}^{T}P_{2,k,l}^{-1}\left({\hat{\varphi }}_{k,l}-\varphi_{k,l}\right)$$

## 6. Conclusions

In this paper, the observability of cooperative positioning system based on two leaders that broadcast their position information alternately and a consistent EKF multi-AUV cooperative positioning algorithm is proposed. Observability analysis results show that the standard EKF has the problem of obtaining forged measurement information along the unobservable direction, which leads to inconsistent state estimation. The algorithm proposed in this paper adds the zero-space vector as the observability constraint, which improves the consistency of the cooperative positioning system. Simulation results show that the NEES expected values of position and heading angle estimations are 4 m and 1.015 rad, respectively, when using the EKF algorithm, and close to 2 m and 1 rad, respectively, when using the consistent EKF algorithm. Therefore, the consistent EKF algorithm obtained the NEES expected values closer to the real expected values, and its estimated performance is better than the EKF algorithm. Moreover, the consistent EKF algorithm improves the positioning accuracy of the follower AUV, keeps the follower AUV synchronized with the leader AUV, and then keeps the formation in the process of travelling. At present, we have completed the simulation of the algorithm, and some experiments will be carried out in the future. In the future research, we will try to improve the robustness of the system by reducing the communication frequency. We will also study the practical implementation problems in real world applications, such as scanning the seabed with a group of AUV, expanding the working scope and improving the working efficiency.

## Figures and Tables

**Figure 1 sensors-22-04563-f001:**
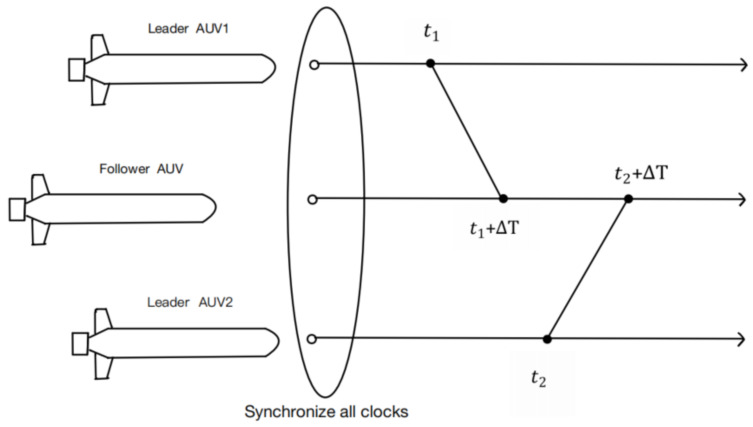
Architecture of multiple AUVs cooperative localization based on two leaders alternately.

**Figure 2 sensors-22-04563-f002:**
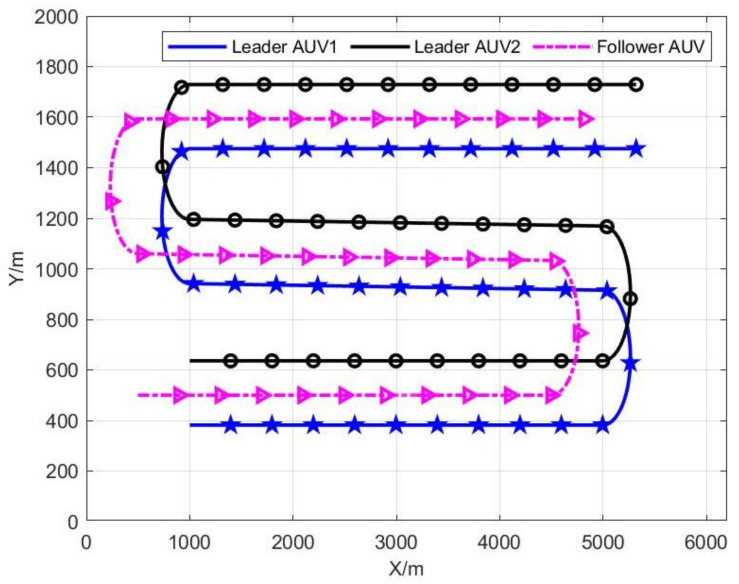
Real trajectories of leader and follower AUVs.

**Figure 3 sensors-22-04563-f003:**
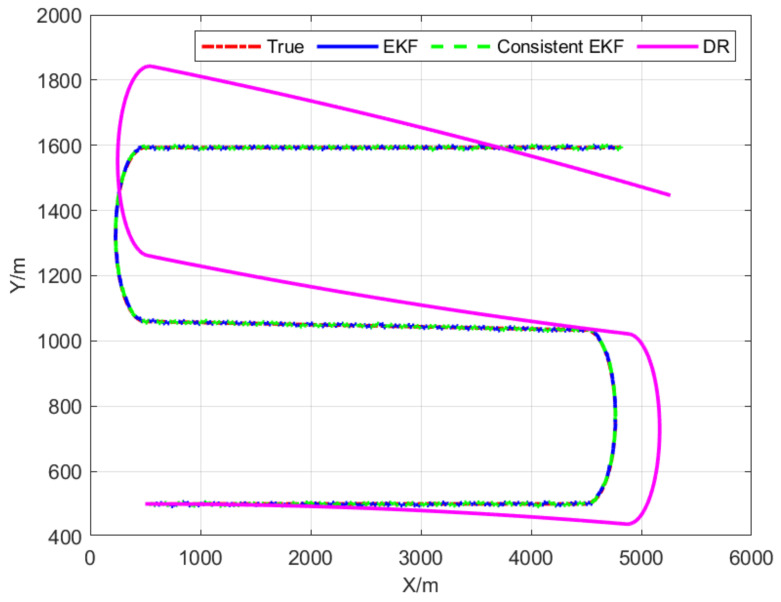
Real and localization trajectories of the follower AUV.

**Figure 4 sensors-22-04563-f004:**
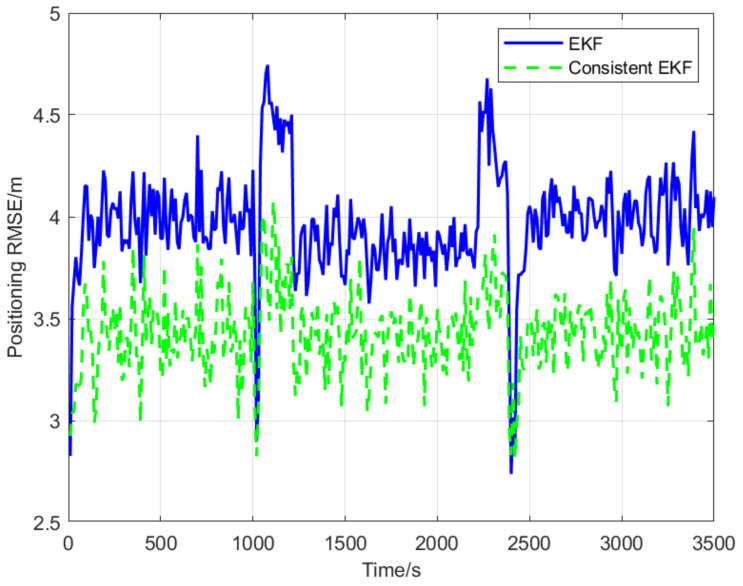
Root mean square errors of position.

**Figure 5 sensors-22-04563-f005:**
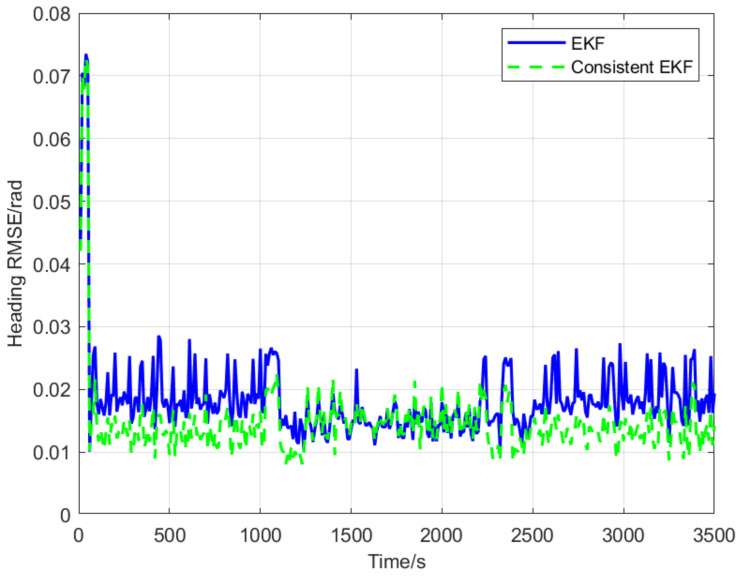
Root mean square errors of heading.

**Figure 6 sensors-22-04563-f006:**
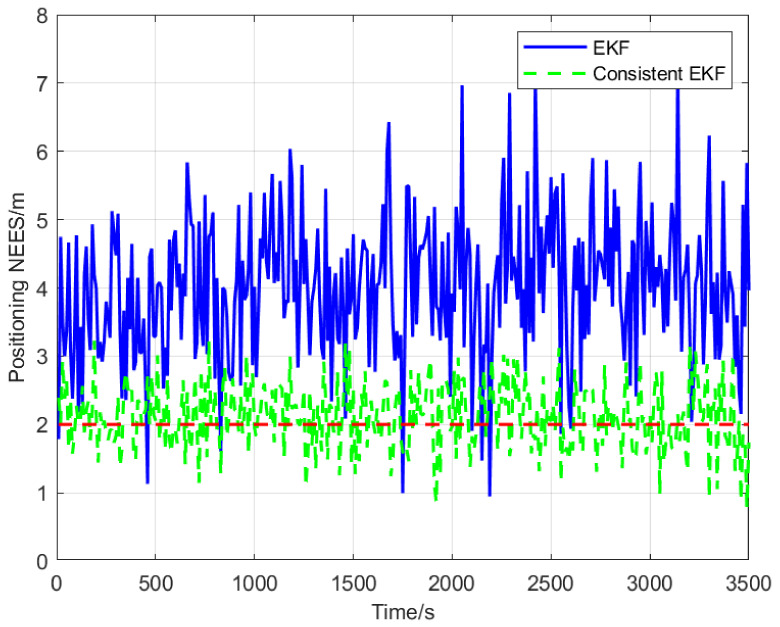
Normalized estimation error squared of position.

**Figure 7 sensors-22-04563-f007:**
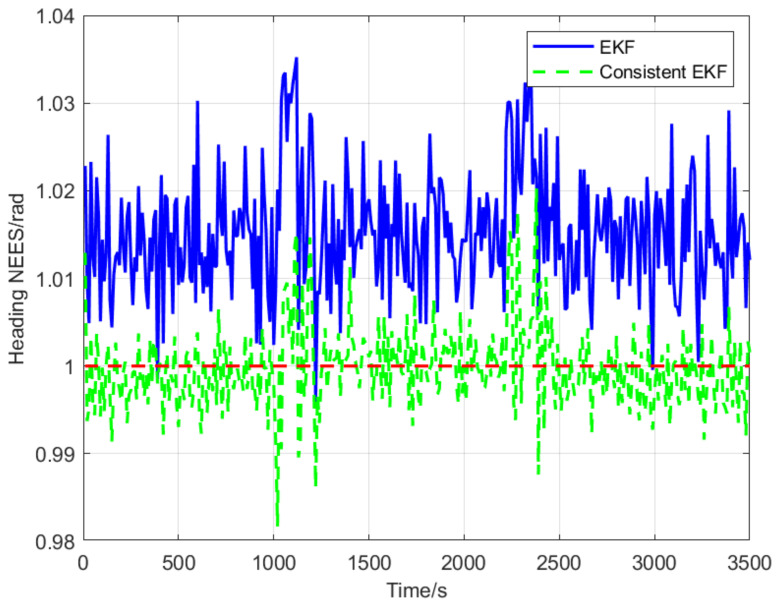
Normalized estimation error squared of heading.

## Data Availability

Not applicable.

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
