# Peer review of "Consistent Extended Kalman Filter-Based Cooperative Localization of Multiple Autonomous Underwater Vehicles"

_sensors, 2022, doi:10.3390/s22124563_

Round 1

Reviewer 2 Report

The manuscript presents a cooperative localisation of multiple autonomous underwater vehicles exploiting an extended Kalman filter.

In general the manuscript is well organised and well written.

The theory of multi autonomous underwater cooperative positioning and the basis concepts related to the standard EKF have been deeply investigate in section 2, section n.3 reports the observability analysis whilst in section 4 a consistent extended Kalman filter is proposed  and a consistency analysis.

In my opinion the paper is very interesting and pleasant to read. Moreover the presented results verify the effectiveness of the proposed strategy.

The conclusion adequately resume the application context and the obtained results.

My concern are as in the following.

I suggest  to improve the introduction section. I expect that before to describe the paper organisation, the authors have to describe the main contributions of the paper.

In this respect the authors state as follows:

To sum up, the main contributions of this work are as follows.

(1) ln this paper, follower AUV ….

(2) According to different ….

(3) In order to improve …

In my opinion, as in the conclusion section, the authors must better highlight and synthesise the motivation and main contribution of the paper.

Furthermore, I suggest to enrich the scientific background by discussing about position and localisation system that jointly exploits EKF and acoustic communication system. 

Then, in order to improve the research background, I suggest to consider and discuss about the following contributions: 

- E. M. Sozer, M. Stojanovic and J. G. Proakis, "Underwater acoustic networks," in IEEE Journal of Oceanic Engineering, vol. 25, no. 1, pp. 72-83, Jan. 2000, doi: 10.1109/48.820738.

- Cario, G.; Casavola, A.; Gagliardi, G.; Lupia, M.; Severino, U.; Bruno, F. Analysis of error sources in underwater localization systems. In Proceedings of the OCEANS, Marseille, France, 17–20 June 2019; pp. 1–6, doi:10.1109/OCEANSE.2019.8867536.

- Ullah, I.; Gao, M.S.; Kamal, M.M.; Khan, Z. A survey on underwater localization, localization techniques and its algorithms.In Proceedings of the 3rd Annual International Conference on Electronics, Electrical Engineering and Information Science (EEEIS),Guangdong, China, 8–10 September 2017.

Author Response

Reply to the Comments and Suggestions of Reviewer #2

1.Reply to the suggestion 1 of reviewer #2:

Comment:I suggest to improve the introduction section. I expect that before to describe the paper organisation, the authors have to describe the main contributions of the paper

In this respect the authors state as follows:

To sum up, the main contributions of this work are as follows.

(1) ln this paper, follower AUV ….

(2) According to different ….

(3) In order to improve …”

In my opinion, as in the conclusion section, the authors must better highlight and synthesise the motivation and main contribution of the paper.”

Reply: Thank you for your suggestions. In the revised version, we clearly explain the contribution and novelty of this article. The details are as follows.

To sum up, the main contributions of this work are as follows.

(1) In this paper, the cooperative localization of heterogeneous AUVs based on underwater acoustic communication is studied. In order to improve the positioning accuracy of the follower AUV, the follower AUV can only alternately obtain the relative range measurement values with the two leading AUVs through one way travel time (OWTT), thus reducing the complexity of acoustic communication. Not only obtain better observability than the single leader with the same communication load, but also avoid changing the formation frequently. In addition, the research results can be easily extended to any number of AUV teams and the acoustic communication load will not increase.

(2) According to different distance measurement information, the observability matrix of the whole system is decomposed into two independent parts. The whole positioning system is observable, two state estimation and decomposition subsystems related to a single leader are observable, and two ideal decomposition subsystems related to a single leader are unobservable. Because the rank of observability matrix of decomposition subsystem calculated by standard linearization estimator (such as EKF) is larger than that calculated by the ideal state value, it will lead to inconsistent estimation of the position state of follower AUV.

(3)   In order to improve the consistency of state estimation, this paper designs a consistent EKF algorithm for multi-AUV cooperative positioning. Because the ideal state value can't be used to calculate the Jacobian matrix, in order to improve the consistency of the standard linearization estimator, the algorithm uses the designed initial zero space vector related to the relative position to construct the constrain conditions of each recursive time step, and then obtain the modified measurement Jacobian matrix under the constraint conditions, and proves that the state propagation Jacobian matrix is not affected by the initial zero-space vector.

Thank you for pointing out this for us and please refer to the Introduction for more details.

2.Reply to the suggestion 2 of reviewer #2:

Comment: Furthermore, I suggest to enrich the scientific background by discussing about position and localisation system that jointly exploits EKF and acoustic communication system.

Then, in order to improve the research background, I suggest to consider and discuss about the following contributions:

- E. M. Sozer, M. Stojanovic and J. G. Proakis, "Underwater acoustic networks," in IEEE Journal of Oceanic Engineering, vol. 25, no. 1, pp. 72-83, Jan. 2000, doi: 10.1109/48.820738.

- Cario, G.; Casavola, A.; Gagliardi, G.; Lupia, M.; Severino, U.; Bruno, F. Analysis of error sources in underwater localization systems. In Proceedings of the OCEANS, Marseille, France, 17–20 June 2019; pp. 1–6, doi:10.1109/OCEANSE.2019.8867536.

- Ullah, I.; Gao, M.S.; Kamal, M.M.; Khan, Z. A survey on underwater localization, localization techniques and its algorithms.In Proceedings of the 3rd Annual International Conference on Electronics, Electrical Engineering and Information Science (EEEIS),Guangdong, China, 8–10 September 2017.

Reply: Thank you for your suggestions. According to the three references provided by you, we have improved the research background of this paper by jointly using EKF and discussing the location and positioning system of acoustic communication system. The specific contents quoted are as follows:

The traditional approach for ocean-bottom monitoring is to deploy oceanographic sensors, record the data, and recover the instruments. This approach creates long lags in receiving the recorded information. In addition, if a failure occurs before recovery, all the data is lost. The most effective solution  is established real-time communication between AUVs through underwater acoustic sensors. Underwater networks can also be used to increase the operation range of AUV’s [3].

The underwater acoustic sensors equipped with the follower AUV alternately acquires the position coordinates of the leader AUV and the acoustic signal of the time of sending each signal. The leader AUV and the follower AUV are clock-synchronized, and one way travel time (OWTT) technology can be used to calculate the distance at different times. Because the actual application environment is known, the sound speed is measured and fixed. If the time of signal transmission and reception and the speed of sound in the medium are known, the distance between the transmitter and the receiver can be calculated [8].

The multi-AUV cooperative positioning system proposed in this paper can be applied in many aspects, such as improve ocean exploration, collection of oceanographic data, ecological applications i-e water quality and biological monitoring [13].

Thank you for pointing out this for us and please refer to the Introduction for more details.

[3] E. M. Sozer, M. Stojanovic and J. G. Proakis, "Underwater acoustic networks," in IEEE Journal of Oceanic Engineering, vol. 25, no. 1, pp. 72-83, Jan. 2000, doi: 10.1109/48.820738.

[8] Cario, G.; Casavola, A.; Gagliardi, G.; Lupia, M.; Severino, U.; Bruno, F. Analysis of error sources in underwater localization systems. In Proceedings of the OCEANS, Marseille, France, 17–20 June 2019; pp. 1–6, doi:10.1109/OCEANSE.2019.8867536.

[13] Ullah, I.; Gao, M.S.; Kamal, M.M.; Khan, Z. A survey on underwater localization, localization techniques and its algorithms.In Proceedings of the 3rd Annual International Conference on Electronics, Electrical Engineering and Information Science (EEEIS),Guangdong, China, 8–10 September 2017.

Reviewer 3 Report

This work presents a Consistent Extended Kalman filter for multi-AUV co-location on cooperative positioning system follower AUV under two leaders alternately broadcasting its position. The positioning results of the follower AUV under dead reckoning, standard EKF and consistent EKF algorithm are simulated and compared with the real trajectory of the follower AUV. The simulation results confirm the suitability of the proposed algorithm.

The provided information is relevant for the knowledge field. Nevertheless, some issues should be addressed before this manuscript could be considered for publication.

1) The rank of observability matrix of decomposition subsystem is calculated by the linearization estimator; Would the results apply for the whole system (nonlinear system)?

2) The depth information of the follower AUV can be obtained by using the depth sensor, this information can be used to turn a 3D system into a 2D for localization; What could be the expected consequences of the commercial pressure sensors noise?

3) For the simulation, the communication frequency of multi-AUV cooperative navigation was selected as 1Hz; Is that feasible considering standard location systems?

4) Detailed simulation information should be provided (hardware, software, configuration, settings).

5) The Conclusion section should include quantitative results, advantages and disadvantages, limitations, and recommendation for real implementations.

Author Response

Reply to the Comments of Reviewer #3

1.Reply to the comment 1 of reviewer #3:

Comment:The rank of observability matrix of decomposition subsystem is calculated by the linearization estimator; Would the results apply for the whole system (nonlinear system)?”

Reply: Thank you for your comments. As one of the most commonly used cooperative location filtering algorithms, the standard EKF can make full use of the statistical information of measurement equation and measurement error to recursively solve the follower AUV state estimation based on the linearization of the nonlinear cooperative location system model. This paper conducts an observability analysis of the cooperative positioning process based on standard EKF. For the linearized multi-AUV cooperative positioning system, the observability of the system is tested by calculating the rank of the local observable matrix.

2.Reply to the comment 2 of reviewer #3:

Comment: The depth information of the follower AUV can be obtained by using the depth sensor, this information can be used to turn a 3D system into a 2D for localization; What could be the expected consequences of the commercial pressure sensors noise?

Reply: Thank you for your comments. The accuracy of commercial pressure sensors is  of full scale, so the AUV navigation depth can be directly measured by the depth sensor. Since the actual depth information can be measured by the depth sensor in real-time and the measurement accuracy of the depth sensor is high, the depth does not need to be considered in the system equations and the practical working environment of an AUV can be simplified to a two-dimensional (2D) space[1].

[1]Sun, C.; Zhang, Y.; Wang, G.; Gao, W. A New Variational Bayesian Adaptive Extended Kalman Filter for Cooperative Navigation. Sensors 2018, 18, 2538. https://doi.org/10.3390/s18082538

3.Reply to the comment 3 of reviewer #3:

Comment: For the simulation, the communication frequency of multi-AUV cooperative navigation was selected as 1Hz; Is that feasible considering standard location systems?

Reply: Thank you for your comments. As the acoustic signal cannot carry much information, it is necessary to minimize the communication frequency under the premise of a stable system. Underwater acoustic communication has a narrow bandwidth, a low data rate, and it is easy to lose packets. In water, the communication rate decreases as the distance between communication devices increases. In reference [29] of this paper, the communication frequency in multi-AUV cooperative navigation was selected as 1Hz. In the process of multi-AUV collaborative work, in addition to the transfer of cooperative navigation information between the AUV, there is also a need to pass some formation control instructions and there may also be a long-distance between the leader AUV and the follower AUV. So, considering standard positioning systems, a communication frequency of 1Hz is feasible.

4.Reply to the comment 4 of reviewer #3:

Comment:Detailed simulation information should be provided (hardware, software, configuration, settings).

Reply: Thank you for your comments. Detailed simulation information is as follows:

Hardware

CPU: AMD Ryzen 7 5800H

RAM: 16GB

Disk: WDC PC SN730 SDBPNTY-512G-1101

Software

MATLAB2020b

Configuration

A follower AUV and two leader AUVs

Settings

Update frequency of distance measurement: 0.2Hz

Covariance of distance measurement:

Forward speed:

Angular velocity:  (straight line); (turn)

Covariance of control input:

the initial position of the follower AUV: (500,500)

the initial positions of the leader AUV1: (1000,382)

the initial positions of the leader AUV2: (1000,636)

5.Reply to the comment 5 of reviewer #3:

Comment:The Conclusion section should include quantitative results, advantages and disadvantages, limitations, and recommendation for real implementations.

Reply: Thank you for your comments.

Quantitative results: Simulation results show that the NEES expected values of position and heading angle estimations are 4 m and 1.015 rad respectively by using the EKF algorithm, and close to 2 m and 1 rad respectively by using the consistent EKF algorithm. Therefore, the consistent EKF algorithm obtained the NEES expected values are closer to the real expected value, and its estimated performance is better than the EKF algorithm.

Advantages: The proposed consistent algorithm applies to the cooperative navigation of UAVs and robots while their motion formation and communication means are under the requirements of our paper.

Disadvantages: At present, we have completed the simulation of the algorithm, and some experiments will be carried out in the future.

Limitations: How to reduce communication frequency.

Recommendation for real implementations: In a large underwater task region, requiring all AUVs to maintain a constant motion formation along the same linear direction is an effective planning strategy to ensure full-region coverage. When a group of AUVs scans the seafloor to obtain information, the proposed algorithm can be used to increase the scope and efficiency of operations.

Round 2

Reviewer 1 Report

The author responses to my comments and edits to the paper have addressed my concerns. I would like to thank the authors for clearly organizing their responses and connecting them to edits in the paper.

Reviewer 2 Report

The authors fulfilled all my concerns. 

Thank you for their efforts in improving the paper.